# Effects of Dietary Zinc Chloride and Zinc Sulfate on Life History Performance and Hemolymph Metabolism of *Spodoptera litura* (Lepidoptera: Noctuidae)

**DOI:** 10.3390/insects15090687

**Published:** 2024-09-11

**Authors:** Jingwei Qi, Zhenzhou Xia, Yang Yang, Chuanren Li, Zailing Wang

**Affiliations:** 1Hubei Engineering Research Center for Pest Forewarning and Management, Institute of Entomology, College of Agriculture, Yangtze University, Jingzhou 434025, China; qijingwei77@163.com (J.Q.); zhenzhou0511@163.com (Z.X.); yangyang102398@163.com (Y.Y.); 13986706558@163.com (C.L.); 2Huanggang Academy of Agricultural Sciences, Huanggang 438000, China

**Keywords:** two-sex life table, *Spodoptera litura*, untargeted metabolomics, heavy metal stress

## Abstract

**Simple Summary:**

Zinc is a vital nutrient required by all living organisms; however, its impact varies based on Zn concentration and chemical form. This study examined the effect of zinc chloride (ZnCl_2_) and zinc sulfate (ZnSO_4_) on the life history performance and hemolymph metabolism of the common moth, *Spodoptera litura*, which is known to damage many crops. We found that, while low levels of ZnCl_2_ benefit the reproduction of *Spodoptera litura*, higher levels of ZnCl_2_ prolong the preadult developmental period and decrease the preadult survival rate. Additionally, dietary ZnSO_4_ exerts a devastating effect on the survival of *S. litura* larvae, even at the lowest concentration. This helps us better understand the effect of the chemical forms and concentrations of zinc on the biological processes and toxicological impacts on insects.

**Abstract:**

Zinc is an essential micronutrient crucial in various biological processes of an organism. However, the effects of zinc vary depending on its chemical form. Therefore, the aim of this study was to conduct a comparative analysis of the life history performances and hemolymph metabolism of *Spodoptera litura* exposed to different concentrations of dietary zinc chloride (ZnCl_2_) and zinc sulfate (ZnSO_4_), utilizing two-sex life tables and untargeted metabolomics. The preadult survival rate of *S. litura* significantly decreased, while the preadult developmental period of *S. litura* was prolonged as the dietary ZnCl_2_ concentration increased. However, the fecundity of *S. litura* at 50 mg/kg dietary ZnCl_2_ was significantly increased. The intrinsic rate of increase (*r*) and the finite rate of increase (λ) in *S. litura* in the control group (CK, no exogenous ZnCl_2_ or ZnSO_4_ added) and with 50 mg/kg dietary ZnCl_2_ were significantly higher than those at 100 mg/kg, 200 mg/kg, and 300 mg/kg. Dietary ZnSO_4_ exerts a devastating effect on the survival of *S. litura*. Even at the lowest concentration of 50 mg/kg dietary ZnSO_4_, only 1% of *S. litura* could complete the entire life cycle. Furthermore, as the dietary ZnSO_4_ concentration increased, the developmental stage achievable by the *S. litura* larvae declined. High-throughput untargeted metabolomics demonstrated that both 100 mg/kg dietary ZnCl_2_ and ZnSO_4_ decreased the hemolymph vitamins levels and increased the vitamin C content, thereby helping *S. litura* larvae to counteract the stress induced by ZnCl_2_ and ZnSO_4_. Simultaneously, dietary ZnCl_2_ obstructed the chitin synthesis pathway in the hemolymph of *S. litura*, thus extending the developmental period of *S. litura* larvae. These results indicate that low concentrations of Zn^2+^ positively impact populations of *S. litura*, but the effectiveness and toxicity of Zn depend on its chemical form and concentration.

## 1. Introduction

Zinc, an essential micronutrient metal, is crucial for signal transduction, DNA replication, transcription, and protein synthesis in organisms, where it performs structural, catalytic, and regulatory functions [1,2,3]. Additionally, zinc competes with heavy metal ions, such as cadmium, for transporter-mediated ion transport; this competition reduces the cellular absorption rate of heavy metal ions, thereby shielding cells from their toxic effects [4,5,6]. Since the 1930s, zinc has been widely acknowledged as a crucial component in the growth processes of both animals and plants; it is routinely added to animal feeds and plant fertilizers as a nutritional supplement to enhance healthy growth and optimize production [7,8]. However, when zinc concentrations surpass an organism’s regulatory capacity, it damages the midgut structure [9], precipitating neuronal apoptosis or necrosis [10], as well as physiological, behavioral, and metabolic dysfunctions [11,12], ultimately diminishing population survival and fecundity [13,14].

Zinc pollution has emerged as a widespread issue globally [15,16,17]. Zhuang et al. (2009) observed that the average zinc concentration in the surface soil of farmland in the Pearl River Delta significantly exceeded the Environmental Quality Standard of soils in China [18]. Similarly, Wu et al. (1996) reported that zinc concentrations in the leaves and stems of cabbage grown in zinc-enriched soil reached values as high as 1297 and 1141.8 mg/kg, respectively [19]. Additionally, zinc deposition in agricultural soil through organic fertilizers (livestock manure) is a significant factor contributing to zinc enrichment in both the soil surface and plants [15,20]. Researchers have demonstrated that the zinc levels in organic waste (pig and cow manure) progressively increase during composting, resulting in zinc concentrations that significantly exceed those of other heavy metals such as lead (Pb), copper (Cu), and nickel (Ni) in the Netherlands, England, Wales, and China [21,22,23,24,25]. In Japan, zinc has been classified as a pollutant of concern and designated as a model toxicant due to its environmental and health impacts [26].

Zinc chloride (ZnCl_2_) and zinc sulfate (ZnSO_4_) are prevalent zinc salts in both daily life and industry [27]. ZnCl_2_, a compound rarely found in nature, is predominantly synthesized industrially. Conversely, ZnSO_4_ is found in a variety of minerals and serves as the primary zinc additive in feeds and fertilizers [28]. The global market for ZnSO_4_, driven by increasing demand in sectors such as agriculture, animal husbandry, industry, food, and pharmaceuticals, has an annual growth rate of 4.5% (https://dataintelo.com/report/zinc-sulfate-market/, accessed on 14 July 2024). Although the market demand and usage of ZnSO_4_ surpass those of ZnCl_2_, researchers predominantly utilize ZnCl_2_ as a zinc source to assess the effects of excessive zinc intake on insect growth, development, reproduction, physiological functions, immune responses, enzymatic reactions, and metabolism [9,12,29,30,31]. Researchers, however, more frequently study the inhibitory effects of high-concentration ZnSO_4_ solutions, which are used as insecticides sprayed on plant leaves, on the growth, development, and survival of herbivores [32,33]. Therefore, given the widespread use of zinc sulfate in agriculture, animal husbandry, industry, and food production, ZnSO_4_ should be considered as a dietary source to assess the impact of zinc pollution on insects.

*Spodoptera litura* (Fabricisu, 1775) (Lepidoptera, Noctuidae), an important polyphagous agricultural pest globally, damages cotton, tobacco, soybeans, corn, and cabbage, among other crops, affecting more than 90 crops [34]. Due to its well-defined artificial diet and ease of laboratory handling, *S. litura* has become an important model organism for researchers studying the effects of heavy metals on insect life history performances, metabolism, ion transport, and more [35,36]. High-throughput untargeted metabolomics is used to detect differential metabolites of organisms under different treatments, thereby elucidating their metabolic processes [37,38,39]. In this study, we aimed to accurately evaluate the effects of 0 mg/kg (CK), 50 mg/kg, 100 mg/kg, 200 mg/kg, and 300 mg/kg dietary ZnSO_4_ and ZnCl_2_ on the growth, development, reproduction, and population dynamics of *S. litura* using a two-sex life table. Concurrently, high-throughput untargeted metabolomics based on liquid chromatography–mass spectrometry (LC-MS/MS) was employed to reveal the regulation of metabolites in the hemolymph of *S. litura* in response to dietary ZnCl_2_ and ZnSO_4_. The results will provide a deeper understanding for the toxic effects of Zn^2+^ and SO_4_^2-^ and offer new insights for the population management of *S. litura*.

## 2. Materials and Methods

### 2.1. Artificial Diet Content and Preparation

The artificial diet weighed 139 g, consisting of 71.94% pure water (100 g) and 28.06% dry matter (39 g). The artificial diet contained the following: soybean flour, corn flour, and wheat germ, each at 10 g; yeast extract powder at 1.2 g; sucrose at 5 g; agar at 1.6 g; vitamin C at 0.6 g; and sorbic acid, multi-vitamin, methyl p-hydroxybenzoate, and additional sorbic acid, each at 0.2 g. According to Jin et al., 2020, the ZnCl_2_ and ZnSO_4_ (Tianjin Guangfu Fine Chemical Research Institute, China) content added to the artificial diets was calculated using the following formula [29]: weight of ZnCl_2_ = Zn concentration of diets × wet weight of diets × 136.315/65.39,(1)
weight of ZnSO_4_ = Zn concentration of diets × wet weight of diets × 161/65.39,(2)
where 136.315, 161, and 65.39 represent the molecular weights of ZnCl_2_, ZnSO_4_, and Zn, respectively. Therefore, in the artificial diet, the 50, 100, 200, and 300 mg/kg Zn^2+^ concentrations correspond to 14.49 mg, 28.98 mg, 57.96 mg, and 86.94 mg of ZnCl_2_, respectively, and to 17.11, 34.22, 68.44, and 102.66 mg of ZnSO_4_, respectively.

Following the methods of Zhang et al. (2016) [40] and Yang et al. (2022) [36], agar was added to boiling pure water (100 mL), and the mixture was continuously heated and stirred until the agar completely dissolved. Subsequently, different amounts of ZnCl_2_ and ZnSO_4_ were dissolved in the solution prior to adding the mixture of soybean flour, corn flour, wheat germ, sucrose, and yeast. The mixture was then continuously heated and stirred for approximately 10 min. Finally, the artificial diet was poured into transparent plastic boxes measuring 15.5 × 11 × 6 cm, after the temperature of the mixture dropped below 60 °C. 

### 2.2. Insects Source and Rearing

*S. litura* were obtained from the College of Life Sciences, Sun Yat-sen University (Guangzhou, China), in September 2018 and fed a standard artificial diet for more than 10 generations. *S. litura* were maintained under constant conditions: 25.0 ± 0.5 °C, 50 ± 10% relative humidity, and a 12 h dark/light photoperiod. A cohort of 100 newly hatched larvae, randomly selected from egg masses laid on the same day, were individually housed in plastic Petri dishes (4 cm diameter, 1 cm height) and fed artificial diets containing varying concentrations of ZnCl_2_ and ZnSO_4_. The survival and molting of *S. litura* larvae were recorded daily to track the developmental stage. The black exoskeleton shed by the *S. litura* specimens indicated that they had developed to the next instar. Additionally, if the *S. litura* larvae did not move when poked with a brush, did not curl up, appeared soft, and their body color became darker, this indicated that they were dead. Upon reaching the prepupal stage, each larva was relocated to an individual compartment in a 32-compartment tray containing sterile sand to facilitate pupation. The key characteristics of the prepupal stage in *S. litura* include a shorter and broader body, the cessation of feeding, dark brown heads, and a darkened, shiny dorsum. Additionally, prepupal-stage *S. litura* larvae secrete mucus to begin constructing a pupal chamber. Following eclosion, male and female moths that developed on the same dietary treatment and emerged simultaneously were paired in plastic cups (4.3 × 6.7 × 7.5 cm) with wax paper to enable oviposition. If the number of emergent females exceeded that of males, or vice versa, additional specimens were supplemented from a mass-reared population maintained under identical conditions to ensure that all *S. litura* adults were mated. Nutrition for the adult moths was provided by a cotton ball soaked in a 10% honey–water solution, which was renewed daily at the base of each cup. Egg masses deposited on the wax paper were collected daily, and individual eggs were counted using a stereo microscope (SMZ745, Nikon, Tokyo, Japan) until the expiration of all female adults. The longevity of all *S. litura* adults under different treatments was recorded. 

### 2.3. Life Table Analysis

The life history performances (larval and pupal developmental times, survival rates, adult longevity, and fecundity) and population parameters (intrinsic rate of increase *r*, finite rate of increase λ, net reproductive rate *R*_0_, and mean generation time *T*) of *S. litura* were based on the two-sex life table theory, as described by Chi and Liu (1985) [41] and Chi (1988) [42], and they were calculated using the TWOSEX-MSChart software Version 2024.07.06 [43]. The bootstrap method with 100,000 resamples, as proposed by Efron and Tibshirani (1993) [44], was utilized to determine the means and standard errors of these parameters. In addition, the bootstrap technique was also employed to compare the significant differences in *S. litura*’s life history parameters and population parameters between the different groups. This study further explored the age–stage survival rate (*s*_*x**j*_), which is the probability that a newborn (age 0, stage 1) survives to age *x* and stage *j* [45]. The age-specific survival rate (*l_x_*), representing the probability that a newborn survives to age *x*, was also analyzed alongside its relationship with *s_xj_*:(3)$$l_{x}=\sum_{j=1}^{k}s_{xj},$$
where *k* is the number of stages. The age-specific fecundity (*f*_*x*_) is defined as the number of eggs produced by adult females at age *x*. The net reproductive rate (*R*_0_) is a key demographic parameter representing the average number of offspring an individual is expected to produce over its lifetime.
(4)$$R_{0}=\sum_{x=0}^{\infty }l_{x}m_{x}$$

The intrinsic rate of increase (*r*), indicative of the maximum transient growth rate which a stable age-structured population can achieve, is calculated employing the iterative bisection method based on the Euler–Lotka equation, as described by Goodman (1982) [46]:(5)$$\sum_{x=0}^{\infty }e^{-r(x+1)}l_{x}m_{x}=1$$

The finite rate of increase (λ), which measures the daily average rate at which a population grows when not limited by environmental constraints, is calculated using the following expression:λ = *e^r^*(6)

The mean generation time (*T*) is defined as the time required for a population to increase to *R*_0_-fold of its size under a stable age–stage distribution. This parameter quantifies the average period necessary for a population to replicate its size according to the net reproductive rate.
*T* = (ln *R*_0_)/*r*(7)

### 2.4. Extraction and Pretreatment of Metabolite

After completing the experiment on the effects of various concentrations of dietary zinc chloride and zinc sulfate on the life history parameters of *S. litura*, we began extracting metabolites from the hemolymph of *S. litura* larvae in August 2023. Sixth instar larvae of *S. litura*, fed on diets with CK, 100 mg/kg dietary ZnCl_2_, and 100 mg/kg dietary ZnSO_4_, were first cleansed using deionized water and subsequently disinfected with 75% alcohol on their body walls. These larvae were then temporarily immobilized by exposure to −10 °C for 10 min. Using a sterile insect needle, the larval body wall was punctured at the mid-abdomen, and hemolymph was collected using microcapillaries. The extracted hemolymph was transferred into 1.5 mL centrifuge tubes pretreated with 0.025% phenylthiocarbamide to prevent blackening reactions and then stored at −80 °C. For each experimental treatment, hemolymph was collected from 80 larvae, and 1 mL of hemolymph was pooled from every 10 larvae to form a sample, with each treatment replicated eight times.

Metabolite pretreatment of the *S. litura* hemolymph for ultraperformance liquid chromatography–tandem mass spectrometry (UHPLC-MS/MS) Agilent Technologies Inc., California, and SUA, analysis was conducted following the protocol described by Cheng (2023) [47]. Briefly, 200 μL of thawed hemolymph was mixed with 800 μL of methanol and agitated for 60 s. This mixture was then centrifuged at 14,000 rpm for 10 min at 4 °C. The resulting supernatant was transferred to a new centrifuge tube and subjected to freeze-drying. The dry sample was reconstituted in 400 µL of a 4 ppm 2-chlorophenylalanine-methanol aqueous solution (1:1, *v*/*v*) at 4 °C and filtered through a 0.22 µm filter membrane into a sampling vial for subsequent HPLC-MS/MS analysis. Additionally, 20 μL from each prepared sample was pooled to create a quality control (QC) sample, used to correct deviations in analytical results and instrumentation errors.

### 2.5. UHPLC-MS/MS Analysis

Metabolites in the hemolymph of *S. litura* were characterized and quantified using UHPLC-MS/MS on a Waters ACQUITY UPLC HSS T3 system. Chromatography was performed using an ACQUITY UPLC HSS T3 column Beijing Yuwei Technology Co., Ltd, Beijing, and China, (150 × 2.1 mm, 1.8 μm, Waters) with the autosampler temperature maintained at 8 °C and the column temperature at 40 °C. An injection volume of 2 μL was used. The mobile phases, composed of formic acid in water (1:1000, *v*/*v*) and formic acid in acetonitrile (1:1000, *v*/*v*), were delivered at a flow rate of 0.25 mL/min. Mass spectrometry was performed using electrospray ionization in both positive (ESI+) and negative ion modes (ESI−) with capillary temperatures set at 325 °C and spray voltages of 3.8 kV (positive) and −2.5 kV (negative). The mass analyzer operated from 81 to 1000 *m*/*z* at a resolution of 70,000 in full scan mode.

Peak area data for hemolymph metabolites from larvae fed diets with CK, 100 mg/kg ZnCl_2_, and 100 mg/kg ZnSO_4_ concentrations were normalized. A total of 501 metabolites annotated in the hemolymph of *S. litura* larvae were used for principal component analysis (PCA) and partial least squares discriminant analysis (PLS-DA). In SIMCA-P v. 14.0, PCA and PLS-DA were employed for unsupervised and supervised data analysis, respectively, aiding in data clustering and preventing model overfitting. Metabolites demonstrating a Variable Importance in the Projection (VIP) > 1, a *p*-value < 0.05 (*t*-test), and a Fold Change (FC) > 1.2 or <0.833 were identified as significantly different across the treatments. Metabolite annotation was conducted using the KEGG database (https://www.genome.jp/kegg/, accessed on 24 July 2024), which provided insights into the biochemical metabolic and signal transduction pathways of the identified differential metabolites.

## 3. Results

### 3.1. Effects of Dietary ZnCl_2_ on the Development and Survival of S. litura

The developmental period of the third instar larvae, fourth instar larvae, fifth instar larvae, and preadults of *S. litura* was significantly extended as the concentration of ZnCl_2_ increased. Compared to the preadult developmental period of *S. litura* at 0 mg/kg dietary ZnCl_2_ (48.5 d), the preadult developmental period of *S. litura* at 100 (57.8 d), 200 mg/kg (59.4 d), and 300 mg/kg (56.4 d) dietary ZnCl_2_ was significantly extended by 9.3 d, 10.9 d, and 7.9 d, respectively. Simultaneously, the preadult survival rate of *S. litura* significantly decreased as the dietary ZnCl_2_ concentration increased (Table 1). Compared to a preadult survival rate of *S. litura* at 0 mg/kg ZnCl_2_ (70.0%), the preadult survival rates at 50 mg/kg (47.9%), 100 mg/kg (33.0%), 200 mg/kg (27.0%), and 300 mg/kg (5.4%) dietary ZnCl_2_ were significantly decreased by 22.1%, 37.0%, 43.0%, and 64.6%, respectively (Table 1). The age–stage survival rates (*s_xj_*) and age-specific survival rate (*l_x_*) for *S. litura* pupae decreased sharply at dietary ZnCl_2_ concentrations of 50 and 100 mg/kg, whereas, for stages L6–8, the age–stage survival rates (*s_xj_*) and age-specific survival rate (*l_x_*) declined rapidly at 200 and 300 mg/kg (Figure 1A–E and Figure 2A).

### 3.2. Effects of Dietary ZnCl_2_ on Longevity and Fecundity of S. litura Adults

There was no significant difference in the longevity of *S. litura* female adults across different dietary ZnCl_2_ concentrations. However, the longevity of males at 300 mg/kg (31.4 d) was significantly longer than at 0 mg/kg (20.9 d) and 50 mg/kg (17.8 d) (Table 1). The fecundity of *S. litura* females at 50 mg/kg (1022.4 egg/female) dietary ZnCl_2_ was significantly higher than that at 0 mg/kg (620.4 eggs/female) and 200 mg/kg (592.9 eggs/female) (Table 1). The oviposition days of *S. litura* females on 100 mg/kg (10.7 d) dietary ZnCl_2_ were significantly longer than those on 0 mg/kg (6.5 d), 50 mg/kg (7.3 d), 200 mg/kg (4.8 d), and 300 mg/kg (6.9 d) dietary ZnCl_2_. The age-specific female fecundity curves (*f_x_*) showed two egg-laying peaks of *S. litura* female at various ZnCl_2_ concentrations (Figure 2). The maximum *f_x_* value for *S. litura* at 50 mg/kg (210 egg/female) and 300 mg/kg (305 eggs/female) was higher than at 0 mg/kg (122.2 eggs/female), 100 mg/kg (172 eggs/female), and 200 mg/kg (157 eggs/female) (Figure 2C).

### 3.3. Effects of Dietary ZnCl_2_ on the Population Parameters of S. litura

The intrinsic rate of increase (*r*) and the finite rate of increase (λ) of *S. litura* significantly decreased with increasing dietary ZnCl_2_ concentrations. There was no significant difference in the *r* and λ of *S. litura* at the CK (*r* = 0.092, λ = 1.0963) and 50 mg/kg (*r* = 0.0985, λ = 1.1035) dietary ZnCl_2_ concentrations. However, these values were significantly higher than those at the 100 (*r* = 0.0728, λ = 1.0755), 200 (*r* = 0.0637, λ = 1.0658), and 300 mg/kg (*r* = 0.0494, λ = 1.0507) mg/kg dietary ZnCl_2_ concentrations. There was no significant difference in the net reproductive rate (*R*_0_) of *S. litura* at the CK (211.05), 50(245.02), and 100 mg/kg (134.46) dietary ZnCl_2_ concentrations, but it was significantly higher than that at the 300 mg/kg (21.16) dietary ZnCl_2_ concentration (Table 2).

### 3.4. Effect of Dietary ZnSO_4_ on Life History Performances

Only 1% of *S. litura* larvae on 50 mg/kg dietary ZnSO_4_ completed the entire life cycle, and those fed higher concentrations of ZnSO_4_ were unable to develop into adults. As the ZnSO_4_ concentration increased, the developmental stages that *S. litura* larvae reached were progressively lower (Figure 1F–I). *S. litura* larvae that were fed a diet containing 100 mg/kg ZnSO_4_ could develop up to the prepupal stage. However, *S. litura* larvae on 200 mg/kg dietary ZnSO_4_ reached only L6–8, and those at 300 mg/kg progressed only to L6 (Figure 1H,I). Dietary ZnSO_4_ resulted in a sharp drop in the survival rate of *S. litura* larvae. Higher concentrations of ZnSO_4_ had no significant effect on the *s_xj_* of early larval stages (L1, L2, L3, and L4). Following the onset of the later larval stages, dietary ZnSO_4_ significantly reduced the *s_xj_* and *l_x_* of *S. litura* larvae (Figure 1F–I and Figure 2B).

### 3.5. Metabolic Profiles of S. litura Fed Diets with Different Treatments

The PCA and PLS-DA models demonstrated complete separation of the CK, 100 mg/kg dietary ZnSO_4_, and 100 mg/kg dietary ZnCl_2_ at a 95% confidence level, suggesting significant differences in the metabolic characteristics between the treatments (Figure 3A,B). Metabolites showing significant differences were identified from 501 annotated hemolymph metabolites by setting VIP > 1.0, FC > 1.2 or FC < 0.833, and *p*-value < 0.05. In the ZnCl_2_ group, 121 metabolites were significantly upregulated, and 199 were noticeably downregulated compared to the CK group (Figure 3C). In the ZnSO_4_ group, 193 metabolites were substantially upregulated, and 90 were noticeably downregulated compared to the CK group (Figure 3D). A total of 118 compounds were remarkably upregulated, and 98 metabolites were notably downregulated in the ZnCl_2_ group compared to the ZnSO_4_ group (Figure 3E).

KEGG pathway enrichment identifies the primary biochemical metabolic and signal transduction pathways associated with differential metabolites. In the comparison between CK and ZnCl_2_, chitin synthesis and the vitamin digestion and absorption pathways were identified as primary biochemical metabolic pathways (Figure 3F), with nine metabolites annotated for the former, six of which showed significant differences (Figure 3H), and ten metabolites annotated for the latter, six of which showed significant differences (Figure 3G). 

### 3.6. Metabolic Pathway of S. litura Fed Diets with Different Treatments

A detailed visual analysis of the metabolic pathways for chitin synthesis is shown in Figure 4. In chitin synthesis, trehalose is gradually converted into glucose-6-phosphate, glucosamine 6-phosphate, N-Acetylglucosamine-6-phosphate, N-Acetyl-glucosamine-phosphate, UDP-N-Acetylglucosamine-phosphate, and chitin. The relative concentrations of trehalose, glucose-6-phosphate, glucosamine 6-phosphate, N-Acetylglucosamine-6-phosphate, N-Acetylglucosamine-phosphate, and UDP-N-Acetylglucosamine-phosphate in the hemolymph of *S. litura* larvae fed ZnCl_2_ were significantly lower than those fed CK and ZnSO_4_. The relative content of glutamine in the hemolymph of *S. litura* larvae fed ZnCl_2_ was significantly higher than among those fed CK and ZnSO_4_ (Figure 4). Additionally, in the vitamin digestion and absorption metabolic pathway, the relative content of vitamin H, vitamin B6, vitamin B2, flavin mononucleotide, and folic acid in the hemolymph decreased significantly in *S. litura* fed ZnCl_2_ and ZnSO_4_, compared to CK. However, the content of vitamin C in the hemolymph did not show a significant increase (Figure 5).

## 4. Discussion

Zinc, serving as an essential trace nutrient, confers multiple biological benefits at low concentrations [48]. Additionally, zinc can shield cells from the toxic effects of various toxins and contribute to the structural and functional maturation of neurons [5,35,49,50]. However, an excessive intake of zinc can prolong the larval development time, reduce the larval survival rates, and disrupt the metabolic processes in the hemolymph, as well as physiological and biochemical functions [30,51]. In our study, the preadult developmental period of *S. litura* fed on diet with 100, 200, and 300 mg/kg ZnCl_2_ concentrations was significantly extended by 9.3 d, 10.9 d, and 7.9 d, respectively, compared to the CK group (Table 1). Insects need to expend energy to carry out the detoxification, storage, or excretion of heavy metals [31,52,53]. For example, heavy metals may generate an oxidative stress response, compelling organisms to synthesize additional peptides and enzymes, such as metallothioneins (MT) and heat shock proteins (Hsps), to mitigate the toxicity of heavy metals, resulting in insufficient energy for insects, thereby adversely affecting their growth, development, and reproduction [54]. Furthermore, excessive zinc binds to the sulfhydryl (-SH) groups in critical proteins (enzymes involved in various metabolic pathways), which leads to reduced ATP synthesis and an ensuing energy crisis in organisms [55]. Consequently, when *S. litura* consumes excessive amounts of ZnCl_2_ from its diet, it diverts more energy from food nutrients to detoxification processes, resulting in decreased body weight and prolonged larval development.

Additionally, the preadult survival rates of *S. litura* at dietary ZnCl_2_ concentrations of 100, 200, and 300 mg/kg significantly decreased by 22.1%, 37.0%, 43.0%, and 64.6%, respectively, compared to the CK group (Table 1). Jin et al. (2020) [29] found that, when *S. litura* was fed ZnCl_2_ at concentrations ranging from 150 to 450 mg/kg, the relative consumption rate (RCR) and approximate digestibility (AD) of the sixth instar larvae increased, leading to rapid zinc accumulation in the body. Moreover, Shu et al. (2012) [9] observed that the zinc content in the midgut of *S. litura* increased with increasing dietary ZnCl_2_ concentrations, leading to the significant formation of metallothioneins (MT) in the midgut and the emergence of numerous electron-dense granules (EDGs) and vacuoles in the cytoplasm of midgut cells. This suggests that, during the period of overeating, *S. litura* rapidly ingests ZnCl_2_ from its diet, failing to excrete excess zinc, resulting in increased mortality in older larvae.

Although feeding *S. litura* high concentrations of ZnCl_2_ resulted in a prolonged preadult developmental period and decreased preadult survival rates, the fecundity of adults fed 50 mg/kg ZnCl_2_ was significantly higher than that of adults fed the CK diet or higher concentrations (200 mg/kg) of ZnCl_2_ (Table 1). Similarly, Shephard et al. (2020) [56] found that Lepidopteran pest (*Pieris rapae*) larvae developing on intermediate concentrations of ZnCl_2_ had higher adult fecundity. Indeed, zinc, as a major structural component of many enzymes and transcription factors, can activate regulatory proteins and increase proliferation and differentiation during vitellogenesis [57]. Additionally, Falchuk and Montorzi (2001) [58] found that vitellogenin (Vg) is a metalloprotein containing zinc and calcium. When Vg is processed into vitellin in the oocyte, zinc is also transported to the ovary via the hemolymph and taken up by the Vg-bound oocyte. Therefore, an appropriate intake of zinc can facilitate the formation of vitellogenin in the eggs of *S. litura*, thereby enhancing their reproductive capacity. However, an excessive intake of zinc can significantly diminish the reproductive capacity of *S. litura*. Shu et al. (2009) [30] demonstrated, using Western blotting and inductively coupled plasma atomic emission spectrometry, that an excessive zinc intake leads to the accumulation of zinc in the ovaries of *S. litura*, thereby disrupting ovarian development, reducing yolk protein content, and ultimately decreasing fecundity. Similarly, Al-Dhafar and Sharaby (2012) [32] also observed that zinc accumulation in yolk granules (vitellogenesis) and follicular epithelial cells disrupts the production of female gametes, thereby causing ovarian malformations and reducing the fecundity of insects.

Surprisingly, even when *S. litura* fed on the highest concentration (300 mg/kg) of ZnCl_2_, 5% of the *S. litura* specimens could complete the entire life cycle. However, only 1% of the *S. litura* specimens fed on a lower concentration (50 mg/kg) of ZnSO_4_ could complete the entire life cycle. Therefore, compared to the effect of ZnCl_2_ at the same concentration on the survival of *S. litura*, we suspect that dietary SO_4_^2−^ might have a devastating effect on the survival of *S. litura*. Kavitha et al. (2012) [59] found that dietary ZnSO_4_ reduced the protein levels in the silk gland and hemolymph by 356% and 181%, respectively. Additionally, sulfate salts, due to their unique insecticidal properties, also lead to insect mortality. For example, FeSO_4_ is often used as an animal insecticide for the control of stick insect pests [60,61,62]. However, it is not clear whether sulfate ions or metal cations cause the death of insects, although Al-Dhafar and Sharaby (2012) [32] found that a 0.566% ZnSO_4_ solution caused vacuolation and contraction of midgut epithelial cells and goblet cells in the distal part of the midgut, contraction of some peritrophic membranes, and shedding of some muscle layers in *Rhynchophorus ferrugineus*. Additionally, Halpern et al. (2002) [63] discovered that CuSO_4_ increased the permeability of the plasma membrane of the midgut cells of *Chironomus luridus*, allowing fluid from the intestinal cavity to freely enter the extracellular space, resulting in death. However, it is difficult for us to determine whether SO_4_^2−^ alone or SO_4_^2−^ and metal cations together negatively affect insects. From our experimental results, we can determine that, compared to chloride metal salts, SO_4_^2−^ has a devastating effect on the survival of insects, which may be an important reason why CuSO_4_ and FeSO_4_ are used as insecticides.

To further explore the regulation of metabolites in the hemolymph of *S. litura* under the influence of ZnCl_2_ and ZnSO_4_ intake, we utilized untargeted metabolomics to analyze the metabolic characteristics and profiles. The relative concentrations of glutamine and various chemicals in the chitin synthesis pathway (Glucose-6-phosphate, Glucosamine-6-phosphate, N-acetylglucosamine-6-phosphate, N-acetylglucosamine-1-phosphate, and UDP-N-acetylglucosamine phosphate) in *S. litura* on dietary ZnCl_2_ were significantly lower than those on CK and dietary ZnSO_4_ (Figure 4). Chitin, a well-known linear homopolymer composed of β-1,4-linked N-acetylglucosamine, is found in the anterior cuticle, trachea, and muscle attachment sites, where it functions alongside proteins and other components to provide structural support for the insect exoskeleton, tracheal system, and digestive tract [64,65,66,67]. The inhibition of chitin synthesis impedes the formation of the insect exoskeleton, disrupts the molting processes, and can ultimately result in death [68,69,70]. However, the inhibition of chitin synthesis is influenced by various factors, including the activities of several enzymes—trehalose (Tre), glucose-6-phosphate isomerase (G6PI), fructose-6-phosphate aminotransferase (GFAT), glucosamine-6-phosphate N-acetyltransferase (GNA), phosphoglucosamine mutase (PAGM), and UDP-N-acetylglucosamine pyrophosphorylase (UAP)—which are involved in converting trehalose into chitin [71,72,73]. For example, Liu et al. (2015) [74] found that cadmium stress can stimulate the strong expression of the GFAT protein in the hepatopancreas, thereby regulating chitin biosynthesis. Another possible factor is the availability of raw materials for chitin synthesis in the hemolymph, such as glycogen and trehalose, which can lead to the blockage of chitin synthesis. The relative contents of various substances of *S. litura* on ZnSO_4_ and CK diets were significantly higher than those on ZnCl_2_, except for glutamine (Figure 4). Glutamine mainly acts as an ammonia donor in the synthesis of chitin and irreversibly converts fructose-6-phosphate into glucosamine-6-phosphate under the catalysis of fructose-6-phosphate amidotransferase [75]. This indicates that glutamine is not consumed but rather accumulates in the hemolymph during the chitin synthesis process [76]. Therefore, *S. litura* expends more energy to resist the effects of Zn^2+^ stress, leading to a reduction in trehalose in the hemolymph, which, in turn, hinders the synthesis of chitin and ultimately results in a prolonged developmental period or the death of the insect.

The formation of free radicals constitutes the primary toxic effect of metals on organisms, as these radicals can induce DNA damage, alter sulfhydryl homeostasis, and promote lipid peroxidation [77]. Numerous vitamins have been demonstrated to mitigate the physiological damage resulting from an excessive intake of metal minerals [78,79,80]. For example, vitamin C and vitamin E, acting as reducing agents, inactivate free radicals, thereby mitigating the oxidative damage caused by heavy metals to organisms and providing antioxidant protection against oxidative stress [81]. Vitamin C (ascorbic acid) has four hydroxyl groups that can bind to metal substances, making vitamin C a metal oxide surface modifier [82]. For example, Farjan et al. (2012) [79] and Garg and Mahajan (1994) [83] have demonstrated that dietary vitamin C can increase antioxidant activities such as catalase and glutathionease. Therefore, fruits and vegetables rich in vitamin C possess antioxidant properties and can reduce the damage caused by heavy metal poisoning [80]. Similarly, vitamin E, a lipid-soluble non-enzymatic antioxidant, inhibits reactive oxygen species (ROS) production, scavenges hydroxyl radicals, and protects cells from lipid oxidation, mitigating metal-induced damage in vitro and in animals with high concentrations of iron, copper, and cadmium [84,85]. For example, Coskun et al. (2020) [78] demonstrated that dietary vitamin E reduces SOD activity and MDA level in *Galleria mellonella* hemolymph, thereby playing a protective role in selenium poisoning.

## 5. Conclusions

High concentrations of dietary ZnCl_2_ (100 mg/kg, 200 mg/kg, and 300 mg/kg) significantly prolonged the preadult period and reduced the preadult survival rate of *S. litura*, whereas a lower concentration of dietary ZnCl_2_ (50 mg/kg) significantly increased the fecundity of *S. litura*. Additionally, dietary ZnSO_4_ exerts a devastating effect on the survival of *S. litura*. Even at the lowest concentration of 50 mg/kg dietary ZnSO_4_, only 1% of *S. litura* was able to complete the entire life cycle. High-throughput untargeted metabolomics demonstrated that both 100 mg/kg dietary ZnCl_2_ and ZnSO_4_ decrease the content of hemolymph vitamins and increase vitamin C levels, thus helping *S. litura* larvae counteract the stress induced by ZnCl_2_ and ZnSO_4_. Simultaneously, dietary ZnCl_2_ obstructs the chitin synthesis pathway in the hemolymph of *S. litura*, thus extending the developmental period of *S. litura* larvae. These results indicate that the effectiveness and toxicity of zinc depend on its chemical form and concentration. Low concentrations of Zn^2+^ positively affect populations of *S. litura*, and high concentrations of Zn^2+^ negatively affect the population performance and the hemolymph metabolism. Additionally, dietary SO_4_^2-^ exerts a devastating effect on the *S. litura* population.

## Figures and Tables

**Figure 1 insects-15-00687-f001:**
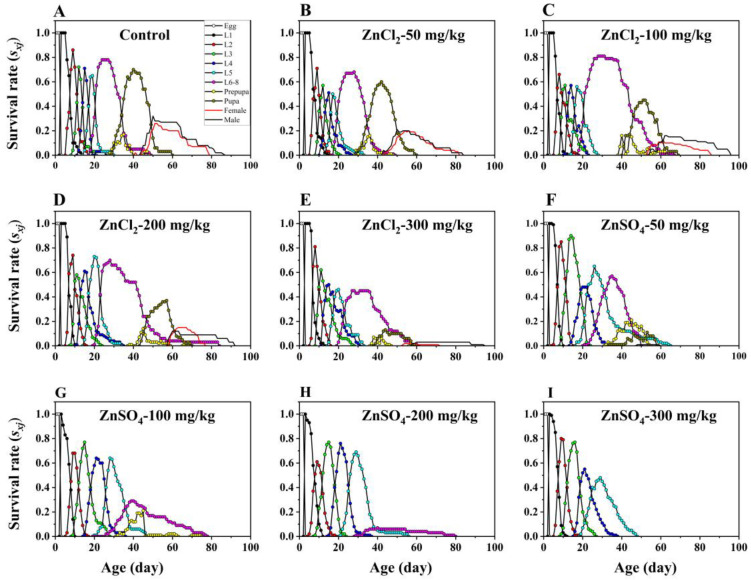
Effect of the CK (**A**), dietary ZnCl_2_ (**B**–**E**) and ZnSO_4_ (**F**–**I**) on the age–stage survival rate (*s_xj_*) of *S. litura*.

**Figure 2 insects-15-00687-f002:**
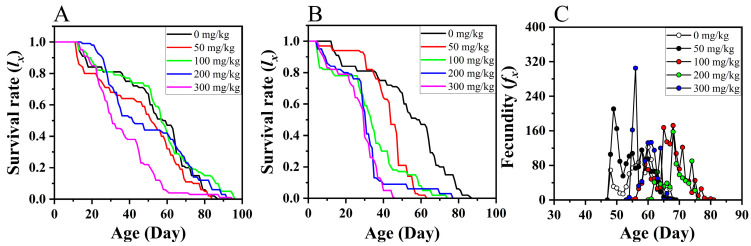
Effect of dietary ZnCl_2_ (**A**) and ZnSO_4_ (**B**) on the age-specific survival rate (*l_*x*_*) and age-specific fecundity (*f_x_*) (**C**) of *S. litura*.

**Figure 3 insects-15-00687-f003:**
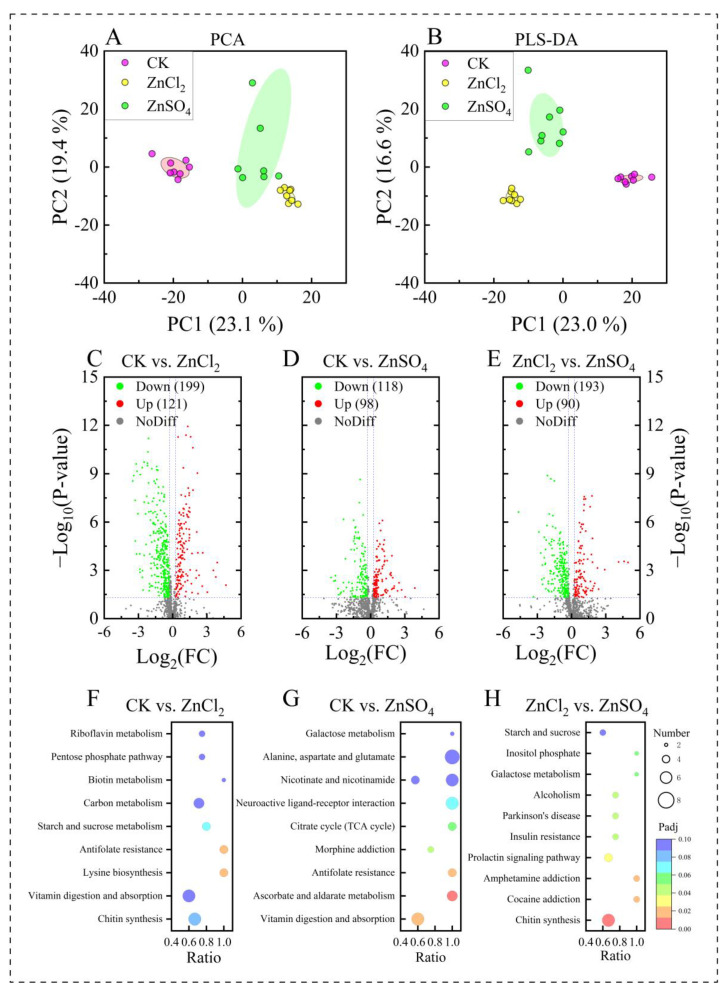
Metabolic profile of *S. litura* reared on diets with CK, 100 mg/kg ZnCl_2_, and 100 mg/kg ZnSO_4_. Principal components analysis, PCA (**A**); partial least squares discrimination analysis, PLS-DA (**B**). The red, yellow, and green shaded regions surrounding the points represent the 95% confidence interval for the CK, dietary ZnCl_2_, and ZnSO_4_ groups, respectively; and volcano plots of changed metabolites of *S. litura* on diets with CK vs. ZnCl_2_ (**C**), CK vs. ZnSO_4_ (**D**), and ZnCl_2_ vs. ZnSO_4_ (**E**). The red, blue, and gray dots represent upregulation, downregulation, and no significant change in the hemolymph metabolites, respectively. “NoDiff” indicates no significant difference. Bubble plot of KEGG pathway enrichment analysis (top 9 enriched KEGG pathways) on diets of CK vs. ZnCl_2_ (**F**), CK vs. ZnSO_4_ (**G**), and ZnCl_2_ vs. ZnSO_4_ (**H**).

**Figure 4 insects-15-00687-f004:**
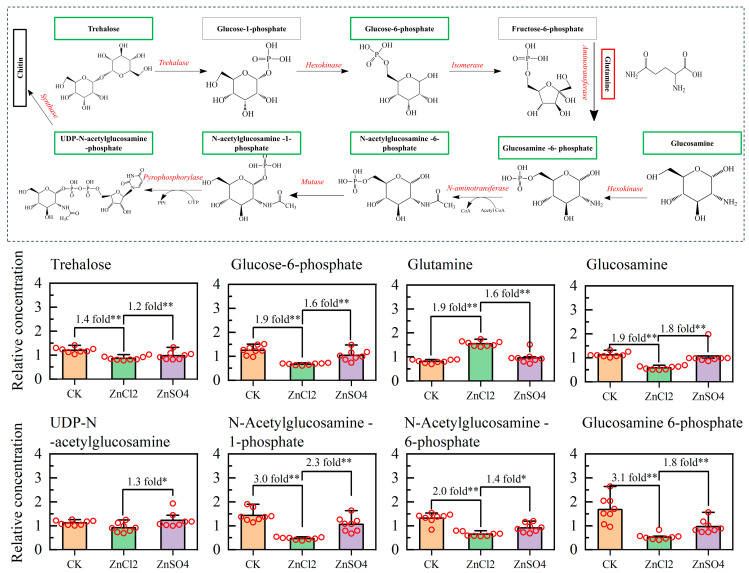
Chitin synthesis pathway and changes in the relative concentrations of metabolites. The arrows indicate the direction of the metabolic pathways. The upper panel shows a series of annotated metabolites in the process of trehalose being gradually converted into chitin in *S. litura* hemolymph. The red borders indicate that the relative concentration of metabolites in *S. litura* hemolymph from specimens reared on dietary ZnCl_2_ is significantly higher compared to the CK and dietary ZnSO_4_ treatments. The green borders indicate that the relative concentration of metabolites in *S. litura* hemolymph from specimens reared on dietary ZnCl_2_ is significantly lower compared to the CK and dietary ZnSO_4_ treatments. The black borders denote that these metabolites were not detected by HPLC-MS/MS. The red circles indicate the relative concentration of the metabolites in each sample. Statistical analyses were conducted using the *t*-test (* *p* < 0.05; ** *p* < 0.01).

**Figure 5 insects-15-00687-f005:**
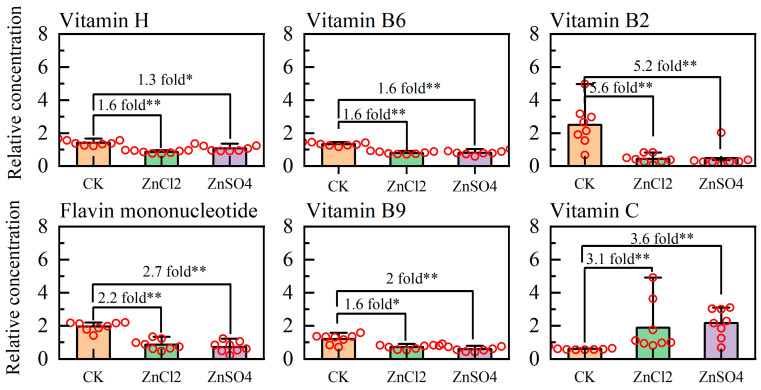
Vitamin digestion and absorption metabolic pathway and changes in the relative concentrations of metabolites. The red circles indicate the relative concentration of the metabolites of each samples. Statistical analyses were conducted using the *t*-test (* *p* < 0.05; ** *p* < 0.01).

**Table 1 insects-15-00687-t001:** The life history parameters of *S. litura* on diets with different ZnCl_2_ concentrations.

Life History Parameters	Concentration (mg/kg)				
	0	50	100	200	300
Egg (d)	3.0 ± 0.00a (100)	3.0 ± 0.00a (100)	3.0 ± 0.00a (100)	3.0 ± 0.00a (100)	3.0 ± 0.00a (100)
L1 (d)	5.0 ± 0.17a (99)	4.8 ± 0.15a (88)	4.8 ± 0.15a (100)	4.7 ± 0.12a (100)	4.7 ± 0.14a (100)
L2 (d)	3.9 ± 0.07a (88)	3.4 ± 0.09bc (83)	3.2 ± 0.08c (100)	3.5 ± 0.09b (100)	3.5 ± 0.13bc (100)
L3 (d)	3.1 ± 0.06d (84)	3.2 ± 0.10cd (80)	3.3 ± 0.08c (94)	3.6 ± 0.10b (100)	4.2 ± 0.18a (91)
L4 (d)	3.2 ± 0.05d (84)	3.3 ± 0.09d (77)	3.5 ± 0.05c (87)	4.1 ± 0.08b (97)	4.5 ± 0.12a (84)
L5 (d)	3.4 ± 0.06c (81)	3.5 ± 0.06c (68)	4.2 ± 0.06b (83)	5.4 ± 0.15a (86)	5.5 ± 0.24a (61)
L6–8 (d)	12.6 ± 0.22d (81)	14.4 ± 0.34c (66)	24.4 ± 0.82a (77)	23.4 ± 0.80a (42)	19.1 ± 1.17b (19)
Prepupa	1.45 ± 0.07c (81)	1.51 ± 0.06bc (66)	1.92 ± 0. 05a (77)	1.95 ± 0.07a (42)	2.01 ± 0.16a (19)
Pupa (d)	13.6 ± 0.21a (76)	13.1 ± 0.13ab (48)	13.0 ± 0.12b (33)	13.0 ± 0.14b (27)	13.4 ± 0.20ab (5)
Preadult (d)	48.5 ± 0.44c (66)	48.8 ± 0.51c (48)	57.8 ± 0.54b (33)	59.4 ± 0.26a (27)	56.4 ± 0.99b (5)
Preadult survival rate (%)	66.0 ± 4.74a (66)	47.9 ± 4.99b (48)	33.0 ± 4.69cd (33)	27.0 ± 4.43d (27)	5.4 ± 2.04e (5)
Oviposition days (d)	6.5 ± 0.75b (34)	7.3 ± 0.85b (24)	10.7 ± 1.17a (15)	4.8 ± 1.35b (15)	6.9 ± 3.04 (2)
Female longevity (d)	14.1 ± 1.69a (34)	15.8 ± 1.83a (24)	15.9 ± 2.32a (15)	12.7 ± 0.92a (15)	12.5 ± 4.93 (2)
Male longevity (d)	20.9 ± 1.92b (32)	17.8 ± 2.11b (24)	23.3 ± 2.58ab (18)	23.2 ± 3.27ab (12)	31.5 ± 2.94a (3)
Fecundity	620.4 ± 137.30b (34)	1022.4 ± 247.13a (24)	895.3 ± 238.30ab (15)	592.9 ± 246.71b (15)	900.7 ± 106.67 (2)

The means and standard errors were calculated using the bootstrap procedure with 100,000 resamples. The means followed by different letters in the same row are significantly different at the 5% significance level. L1 (first instar); L2 (second instar); L3 (third instar); L4 (fourth instar), L5 (fifth instar); and L6–8 (sixth instar to the eighth instar). The sample sizes for oviposition days, female longevity, and fecundity in the 300 mg/kg Zn group were only *N* = 2, indicating an insufficient power to detect meaningful differences. Therefore, the oviposition days, female longevity, and fecundity in the 300 mg/kg dietary ZnCl_2_ group were excluded from the analyses.

**Table 2 insects-15-00687-t002:** Population parameters of *S. litura* specimens reared on diets with different ZnCl_2_ concentrations.

Concentration (mg/kg)	Population Parameters			
	*r* (d^−1^)	λ (d^−1^)	*R*_0_ (Offspring/Individual)	*T* (d)
0	0.0920 ± 0.0058a	1.0963 ± 0.0063a	211.05 ± 54.74a	57.88 ± 1.30b
50	0.0985 ± 0.0065a	1.1035 ± 0.0071a	245.02 ± 72.82a	55.40 ± 1.19b
100	0.0728 ± 0.0059bc	1.0755 ± 0.0064bc	134.46 ± 47.13a	66.39 ± 1.43a
200	0.0637 ± 0.0094cd	1.0658 ± 0.0100cd	88.99 ± 41.63ab	68.18 ± 1.23a
300	0.0494 ± 0.0095d	1.0507 ± 0.0100d	21.16 ± 11.52b	58.99 ± 1.82b

Means in the same row followed by different letters are significantly different at the 5% significance level. *r* (intrinsic rate of increase), λ (finite rate of increase), *R*_0_ (net reproductive rate), and *T* (mean generation time).

## Data Availability

All data generated or analyzed during this study are included in this published article.

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
