# Peer review of "Effects of Dietary Zinc Chloride and Zinc Sulfate on Life History Performance and Hemolymph Metabolism of Spodoptera litura (Lepidoptera: Noctuidae)"

_insects, 2024, doi:10.3390/insects15090687_

Round 1
Reviewer 1 Report
Comments and Suggestions for Authors
This is an informative dose-response study of how two zinc-containing compounds affect life history and metabolic parameters of a common crop pest insect. For the most part, the manuscript is presented in a straightforward way, but I do have concerns about experimental design (see comments below about Lines 130 – 132 and 132 – 134) and the statistical analyses (see comments about Table 1 and Figure 4). Additionally, Figures 4 and 5 require significant clarification (see comments below).
Specific comments:
There are many grammatical errors throughout the manuscript that need to be addressed before publication. A key example is the title, which could be corrected to something like the following: “Effects of dietary zinc chloride and zinc sulphate on life history performance and hemolymph metabolism of Spodoptera litura”.
Line 17: there should be a comma after the word “concentration”.
Line 18: It’s unclear to me how this particular study provides information on how to manage zinc in the environment, since it is just testing for toxicological and metabolomic effects of zinc compounds in a single insect species. This sentence should be revised to better reflect the scope of the study and what the results actually imply.
Line 111: were actual dietary Zn concentrations in each dietary treatment confirmed with a technique such as mass spectrometry? Otherwise, how can the authors be sure the actual dietary Zn concentrations reflect the concentrations they were aiming for?
Line 122: Please provide more details on the history of the study population.
Were they captured directly for the wild or were they from an inbred lab colony? If from a lab colony, then for approximately how many generations have they been reared under lab conditions before the study?
Line `124 – 125: Were there N = 100 larvae in the entire experiment or 100 larvae in each treatment group? Please clarify this and clearly stage how many larvae were reared in each Zn treatment group for each of the two experiments.
Line 129: Please explain how “maturity” was defined in this context, since pre-pupal larvae are technically not mature. Does this refer to when they reached a certain instar of larval development?
Line 130: The word “exclusion” should be replaced with “eclosion”.
Line 130 – 132: Did each mating pair consist of a male and female that developed on the same dietary treatment? Please clarify. Also, note that this design is not ideal given that one cannot infer whether any potential effects of treatment on reproductive parameters are due to females or males being affected by the larval zinc treatment. A more ideal experiment would be to pair Zn-reared females with control males to clearly test effects of Zn on female reproductive traits.
Line 132 – 134: The fact that some mating pairs indeed contained males or females from a separate “mass-reared” population introduces an additional problem for this reproductive experiment, as some mating pairs (I assume) consist of males and females from identical larval treatments while other mating pairs consist of males and females reared under different larval dietary conditions. Was the number of “mass-reared” individuals introduced to the experiment relatively balanced across dietary treatments? If so, the authors can control for this in their statistical analyses and see whether males or females coming from different rearing conditions had any effect on reproductive trait outcomes.
Line 136 – 138: Please explain how eggs were “quantified”. Does this just mean individual eggs were counted?
Line 166: Was it possible to differentiate male from female larvae at this stage? If so, please state if this was done.
Line 205: please provide a citation for the KEGG database.
Line 225: Was lifespan measured only for males and females that had been mated or was lifespan measured for some individuals that were not mated? This is important for the study design since engagement in mating and/or reproductive effort is well known to affect lifespan in insects.
Table 1: It is never explained in the Methods which statistical technique was used to test for treatment differences for each life history trait, even though there are letters to indicate significant differences, which I assume are derived from some kind of post-hoc test. Please state in the methods how these data were analyzed.
An additional question about Table 1: the sample sizes for Oviposition Days, Female longevity, and Fecundity in the 300 mg/kg Zn group appear to be only N = 2, which would mean there is not enough power to detect meaningful differences here. These groups should be excluded from the analyses and it should be explained in the Methods that not enough individuals survived to conduct appropriate statistics on them.
Table 1: please make sure longevity differences were analyzed using proper survival analysis techniques (e.g., Cox regression model), and report this in the Methods.
Figure 3: It would be easier to read this figure if each individual panel had a label showing which comparison is being made.
Figure 4: In the upper pathway panel, the authors use different colors to indicate significant differences, but it is not clear what these significant differences are relative to. This needs to be better explained in the figure caption. Specifically, what was the significant upregulation or downregulation relative to for these analyses?
Figure 4: It is stated that t-tests are being applied, but it is not clear that this type of analysis is appropriate. It looks like for each metabolite a t-test is being used to compare ZnCl2 to CK and then ZNSO4 to ZNCL2. Wouldn’t the more appropriate tests be to compare both ZnCl2 and ZNSO4 to CK? This can be done using an ANOVA with a follow-up planned comparison post-hoc test comparing each group to the control.
Figure 5: Is there an upper panel on this figure that is missing? I do not see a vitamin digestion and absorption metabolic pathway, just the figure showing concentration changes for the 6 metabolites.
Lines 359 – 376: An additional study by Shephard et al. 2020 using another Lepidopteran pest (Pieris rapae) also found that larvae developing on intermediate ZnCl2 concentrations had higher adult fecundity. This study should be mentioned or cited here in the Discussion:
Shephard, A. M., Mitchell, T. S., Henry, S. B., Oberhauser, K. S., Kobiela, M. E., & Snell‐Rood, E. C. (2020). Assessing zinc tolerance in two butterfly species: consequences for conservation in polluted environments. Insect Conservation and Diversity, 13(2), 201-210.
Comments on the Quality of English Language
There are many grammatical errors throughout the manuscript that need to be addressed before publication. A key example is the title, which could be corrected to something like the following: “Effects of dietary zinc chloride and zinc sulphate on life history performance and hemolymph metabolism of Spodoptera litura”.
Reviewer 2 Report
Comments and Suggestions for Authors
This study investigates the impact of zinc at different concentrations on the life history of Spodoptera litura. The larvae were fed diets containing 0 mg/kg (control group), 50 mg/kg, 100 mg/kg, 200 mg/kg, and 300 mg/kg of ZnSO4 and ZnCl2 to assess their effects on growth, development, reproduction, and population dynamics. Additionally, the hemolymph of S. litura was analyzed to explore the effects of zinc on insect physiology and biochemistry.
The presentation of Table 1 does not include the data for APOP and TPOP, yet there are textual explanations in the footnotes. Please review and confirm.
Please confirm that "row" in LINE 264 should be replaced with "column."
Throughout the text, please ensure that ZnSO₄ and ZnCl₂ are correctly formatted with subscript "4" and "2" respectively. Currently, there are multiple instances where the subscript is not properly displayed.
Reviewer 3 Report
Comments and Suggestions for Authors
The authors report in this manuscript the dose-dependent effects of Zn on Spodoptera litura, being differentiated according to its anion counterparts, of which sounds interesting for entomologists. My decision is: this manuscript can be published in the journal after moderate revision. A big problem in this manuscript is the poor description of the statistical analyses which the authors used in this study. The authors should explain what statistical methods were used for what comparisons and also provide their statistics, dfs and probabilities in Results. Also, the authors are suggested to ask a native English speaker to correct English in the revised manuscript before resubmission. Comments below might be available for revision.
Title
The taxonomical group of the species should be referred to: e.g. the armyworm, Spodoptera litura.
L28 CK
What does CK mean? If it indicates some ions, this should be referred to. CK for the abbreviation of control is hard to understand.
Keywords
Italicise the specific name and the common name of this species, armyworm, may be added.
L83
The author, (F.), of the species should be referred to when the species is referred to first in the manuscript.
L88
"U" in untargeted should be written in small character.
L122
The date when the samples were obtained must be referred to.
L128
"Systematically recorded" cannot be understood. Describe clearly and concisely how they were recorded and how different they were from ordinary visual observations.
L129
Define the "maturity". When were the insects determined as "matured" with what morphological characteristics.
L144
Explain the sample size and how resampling was carried out for the bootstrapping.
L167
Refer to the date when the experiment started.
L167, 176
Italicise the specific name.
L187
When was the quantification performed? Refer to the date.
L210
Mention in Materials and Methods how these instars were obtained and how many.
L211
Which test was used for the significance? Provide the statistics and dfs in the test.
L238-242
Refer to the test for the multi-comparison.
L273
Describe in Materials and Methods in detail how the PCA was conducted with what variables. Also mention here what variables PC1 and PC2 represent respectively.
L273
The control should be written as "CK" (as in L28).
L303
The specific name should be in stand characters, if the title is italicised.
L323
No descriptions on the t-test (not "T-test"). Mention how the test was applied to what samples.
L382
Other words, such as adverse, negative or reducible, may be considered instead "devastating", which may be an overstatement.
Comments on the Quality of English LanguageThe issues on the descriptions of statistical analyses and English check must be appropriately treated in the revised manuscript.
Round 2
Reviewer 1 Report
Comments and Suggestions for Authors
The manuscript has been much improved by the revisions. However, I still do not fully agree with some of the statistical analyses, particularly regarding the lifespan and metabolomics analyses. It is not clear how the "bootstrap method with 100,000 resamples" could be used to analyze significant differences between male and female longevity. As the authors suggest in their reply, a log-rank test would be more appropriate.
Regarding the multiple t-tests used to analyze the metabolomics data, consider adjusting p-values using a conservative correction method such as the Bonferroni correction method.
The title of the manuscript still has some grammatical issues. I would suggest revising to something like the following: "Effects of dietary zinc chloride and zinc sulphate on life history performance and hemolymph metabolism of Spodoptera litura (Lepidoptera: Noctuidae)."
Comments on the Quality of English LanguageThere are still grammatical issues that should be addressed before publication.
